# Obesity, Sex, Snoring and Severity of OSA in a First Nation Community in Saskatchewan, Canada

**James A. Dosman** [1,2,*], **Chandima P. Karunanayake** [1,*], **Mark Fenton** [3], **Vivian R. Ramsden** [4], **Jeremy Seeseequasis** [5], **Robert Skomro** [3], **Shelley Kirychuk** [1,2], **Donna C. Rennie** [6], **Kathleen McMullin** [1], **Brooke P. Russell** [1], **Niels Koehncke** [1,2], **Sylvia Abonyi** [7], **Malcolm King** [7] and **Punam Pahwa** [1,7]

1   Canadian Centre for Health and Safety in Agriculture, University of Saskatchewan, 104 Clinic Place, Saskatoon, SK S7N 2Z4, Canada; shelley.kirychuk@usask.ca (S.K.); kathleen.mcmullin@usask.ca (K.M.); bpr053@mail.usask.ca (B.P.R.); niels.koehncke@usask.ca (N.K.); pup165@mail.usask.ca (P.P.)
2   Department of Medicine, University of Saskatchewan, 103 Hospital Drive, Saskatoon, SK S7N 0W8, Canada
3   Respiratory Research Centre, College of Medicine, University of Saskatchewan, 104 Clinic Place, Saskatoon, SK S7N 2Z4, Canada; mef132@mail.usask.ca (M.F.); r.skomro@usask.ca (R.S.)
4   Department of Academic Family Medicine, University of Saskatchewan, West Winds Primary Health Centre, 3311 Fairlight Drive, Saskatoon, SK S7M 3Y5, Canada; viv.ramsden@usask.ca
5   Community A, P.O. Box 96, Duck Lake, SK S0K 1J0, Canada; jccquasis@willowcreehealth.com
6   College of Nursing, University of Saskatchewan, 104 Clinic Place, Saskatoon, SK S7N 2Z4, Canada; donna.rennie@usask.ca
7   Department of Community Health & Epidemiology, College of Medicine, University of Saskatchewan, 107 Wiggins Road, Saskatoon, SK S7N 5E5, Canada; sya277@mail.usask.ca (S.A.); malcolm.king@usask.ca (M.K.)
*   Correspondence: james.dosman@usask.ca (J.A.D.); cpk646@mail.usask.ca (C.P.K.); Tel.: +1-306-966-1475 (J.A.D.); +1-306-966-1647 (C.P.K.)

**Abstract:** Sleep disorders have been related to body weight, social conditions, and a number of comorbidities. These include high blood pressure and type 2 diabetes, both of which are prevalent in the First Nations communities. We explored relationships between obstructive sleep apnea (OSA) and risk factors including social, environmental, and individual circumstances. An interviewer-administered survey was conducted with adult participants in 2018–2019 in a First Nations community in Saskatchewan, Canada. The survey collected information on demographic variables, individual and contextual determinants of sleep health, and objective clinical measurements. The presence of OSA was defined as an apnea–hypopnea index (AHI) $\geq 5$. Multiple ordinal logistic regression analysis was conducted to examine relationships between the severity of OSA and potential risk factors. In addition to the survey, 233 men and women participated in a Level 3 one-night home sleep test. Of those, 105 (45.1%) participants were reported to have obstructive sleep apnea (AHI $\geq$ 5). Mild and moderately severe OSA (AHI $\geq$ 5 to <30) was present in 39.9% and severe OSA (AHI $\geq$ 30) was identified in 5.2% of participants. Being male, being obese, and snoring loudly were significantly associated with severity of OSA. The severity of OSA in one First Nation appears relatively common and may be related to mainly individual factors such as loud snoring, obesity, and sex.

**Keywords:** obstructive sleep apnea; apnea-hypopnea index; First Nations; sex; obesity; loud snoring

## 1. Introduction

Sleep disorders are common in the general Canadian population including those with obstructive sleep apnea (OSA) [1–5]; however, prevalence rates in Indigenous communities are not currently available [6]. Indigenous peoples represent 4.9% of the population of Canada [7]. It is possible that the prevalence of sleep apnea among the Indigenous populations in Canada is high, similar to other health outcomes reported in the literature [8,9].

We have previously reported high rates of snoring and daytime sleepiness in two Cree First Nation communities [10,11] which was suggestive of high rates of OSA. The snoring

that was observed in the communities was related to age, sex, body mass index (BMI), non-traditional use of tobacco (cigarette smoking), sinus trouble and home dampness and mold [10]. Daytime sleepiness among residents of the communities was related to respiratory comorbidities and indices of poverty [11]. Sleep deprivation among community residents who participated in our project was related to housing conditions [12]. Studies in the USA have indicated higher rates of sleep disorders among American Indians compared with other ethnic groups [13,14]. Young et al. found that increased risk for elevated apnea/hypopnea index (AHI ≥ 15) among American Indians could be explained by increased body habitus measurements (height, weight, hip circumference) [15]. Mihaere et al. reported an increased risk ratio for snoring among New Zealand Māori people versus non-Māori populations and a higher prevalence of OSA that appeared attributable to body habitus [16]. Froese et al. reported that snoring in three Indigenous Nations in British Columbia was related to sleepiness [17]. Enhanced snoring risk among New Zealand's Māori people was also related to sleepiness [18].

Socio-economic factors have been associated with sleep disordered breathing [19,20]. Reasons for this could be conditions such as obesity, alcohol and tobacco use, comorbid medical conditions, and barriers to diagnosis; and less than optimal ongoing care associated with low socio-economic status. Studies undertaken in high-income countries have shown a higher prevalence of OSA and sleep conditions in Indigenous and minority groups [21]. The Māori people of New Zealand have been reported to have higher prevalence rates of insomnia, OSA symptoms and excessive sleepiness, and poorer adherence to treatment for sleep disorders compared to non-Māori people [16,18,21–23]. Frequently, Indigenous peoples in Canada, have a low family income, low educational attainment, and inadequate housing (houses with dampness and/or mold) [24–26]. In addition, it has been estimated that 28% of First Nations peoples living on-reserve live in crowded housing conditions with more than one person per room, compared to 4% of non-Indigenous peoples in Canada [27].

Comparing the influence of proximate social determinants on sleep health with other Canadian peoples, there have been, until recently, structural disadvantages for Indigenous peoples as an in-hospital polysomnography was needed. For example, First Nations peoples living anywhere in Saskatchewan previously needed to qualify for treatment under a Federal health insurance scheme, Non-Insured Health Benefits (NIHB), by having an in-hospital sleep test done resulting in treatment delays. In comparison, other residents of Saskatchewan could qualify for treatment based on the one-night at-home simplified "Level 3" sleep test, while NIHB until recently required diagnosis in a sleep laboratory [6,28]. This policy was changed in 2019 as a result of involvement of researchers and physicians who were part of our previous project [28,29]. Now the NIHB program provides coverage for Continuous Positive Airway Pressure therapy (CPAP) devices prescribed by a health professional such as a physician using the Level 3 one-night home sleep test [30] where medically appropriate. It has been shown that an at-home one-night sleep test is adequate for the diagnosis and treatment of moderate to severe OSA [31,32]. This paper examines the prevalence of OSA including a one-night sleep test at-home to identify the main risk factors for OSA in one First Nation community in Saskatchewan, Canada.

## 2. Results

Two hundred and thirty-three individuals participated in the project. Of these, 98 (42.1%) were males and 135 (57.9%) were females. Mean age was 38 years (SD = ±15 years) and mean body mass index was 31 Kg/m$^2$ (SD = ±8 Kg/m$^2$). The mean duration for Apnea Link evaluation was 356 min (SD = ±135 min). Among those, 105 (45.1%) had an AHI index ≥5, which is mild to severe OSA, and 28 (12.1%) had a moderate to severe OSA (AHI ≥ 15). The proportion of AHI index ≥ 5 in males was 51.0% and in females was 40.7%. The prevalence of severe OSA was more than two times higher in males than in females. This is shown in Figure 1.

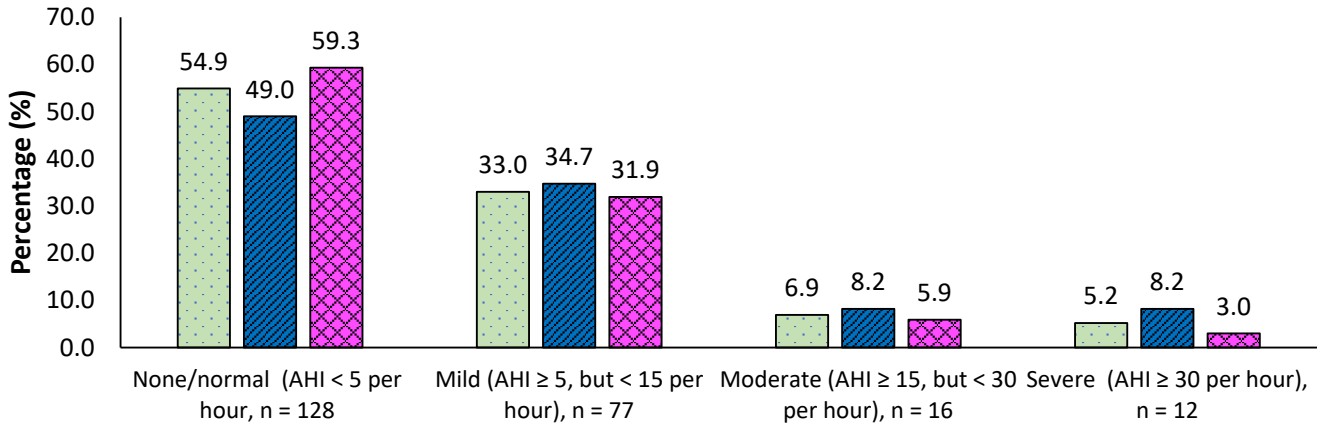

**Figure 1.** Prevalence of Severity of OSA based on Apnea Hypopnea Index (AHI).

Table 1 depicts the association between severity of OSA and selected risk factors. There was a higher proportion of males (16.3%) in the moderate to severe OSA category compared to females. Another interesting observation was that 12.9% of moderate to severe OSA participants were in the 18–39 years age group. The prevalence of obesity was 49.4%. In this study, 38% of males and 58% of females were defined as being obese. A high proportion of individuals reported that they did not have enough money left at the end of the month, and the majority of participants were currently engaged in non-traditional use of tobacco. More than half (53.6%) of the moderate to severe OSA group were current smokers. Those individuals identified as being obese were more likely than those in 'the other' group to detect a significantly higher proportion of mild OSA; as well as moderate to severe OSA compared to the normal group (Table 1). Participants who snored loudly were more likely than those who did not to have a significantly higher proportion of moderate to severe OSA compared to those in the normal group. Associations between socio-economic factors including housing conditions and severity of OSA were not statistically significant, likely because these factors were pervasive throughout the community.

**Table 1.** Association between OSA and selected risk factors (n = 233).

| Variable | Overall n (%) | OSA n (%) | | | *p* Value |
| --- | --- | --- | --- | --- | --- |
| | | Normal (AHI < 5) | Mild OSA (AHI 5–14.99) | Moderate to Severe OSA (AHI ≥ 15) | |
| Sex | | | | | |
| Male | 98 (42.1) | 48 (49.0) | 34 (34.7) | 16 (16.3) | 0.146 |
| Female | 135 (57.9) | 80 (59.2) | 43 (31.9) | 12 (8.9) | |
| Age group, in years | | | | | |
| 18–39 | 140 (60.1) | 80 (57.1) | 42 (30.0) | 18 (12.9) | 0.491 |
| 40–49 | 35 (15.0) | 15 (42.8) | 17 (48.6) | 3 (8.6) | |
| 50–59 | 34 (14.6) | 21 (61.8) | 9 (26.5) | 4 (11.7) | |
| 60 and older | 24 (10.3) | 12 (50.0) | 9 (37.5) | 3 (12.5) | |
| Body Mass Index | | | | | |
| Obese | 115 (49.4) | 48 (41.7) | 48 (41.7) | 19 (16.5) | <0.0001 |
| Other (Normal/overweight) | 118 (50.6) | 80 (67.8) | 29 (24.6) | 9 (7.6) | |

**Table 1.** *Cont.*

| Variable | Overall n (%) | OSA n (%) | | | p Value |
| --- | --- | --- | --- | --- | --- |
| | | Normal (AHI < 5) | Mild OSA (AHI 5–14.99) | Moderate to Severe OSA (AHI ≥ 15) | |
| Smoking Status * | | | | | |
| Current smoker | 161 (70.3) | 90 (55.9) | 56 (34.8) | 15 (9.3) | 0.252 |
| Ex-smoker | 33 (14.4) | 16 (48.5) | 11 (33.3) | 6 (18.2) | |
| Never smoker | 35 (15.3) | 20 (57.1) | 8 (22.9) | 7 (20.0) | |
| Excessive Daytime Sleepiness * | | | | | |
| Yes | 39 (17.3) | 19 (48.7) | 13 (33.3) | 7 (17.9) | 0.407 |
| No | 187 (82.7) | 106 (56.7) | 61 (32.6) | 20 (10.7) | |
| Loud Snoring * | | | | | |
| Yes | 62 (28.8) | 25 (40.3) | 22 (35.5) | 15 (24.2) | 0.001 |
| No | 153 (71.2) | 92 (60.1) | 51 (33.3) | 10 (6.5) | |
| Money left at the end of the month * | | | | | |
| Some money | 47 (20.5) | 26 (55.3) | 17 (36.2) | 4 (8.5) | 0.865 |
| Just enough money | 39 (17.0) | 22 (56.4) | 13 (33.3) | 4 (10.3) | |
| Not enough money | 143 (62.5) | 78 (54.5) | 45 (31.5) | 20 (14.0) | |
| Crowding Index * | | | | | |
| >1 persons/bedrooms | 159 (68.8) | 96 (60.4) | 45 (28.3) | 18 (11.3) | 0.072 |
| ≤1 persons/bedrooms | 72 (31.2) | 32 (44.4) | 30 (41.7) | 10 (13.9) | |
| Damage caused by dampness * | | | | | |
| Yes | 127 (55.2) | 69 (54.3) | 40 (31.5) | 18 (14.2) | 0.432 |
| No | 103 (44.8) | 58 (56.3) | 36 (35.0) | 9 (8.7) | |
| Moldy or musty smell * | | | | | |
| Yes | 112 (48.5) | 62 (55.4) | 38 (33.9) | 12 (10.7) | 0.806 |
| No | 119 (51.5) | 65 (54.6) | 38 (31.9) | 16 (13.4) | |
| Signs of mold in house * | | | | | |
| Yes | 116 (50.7) | 60 (51.7) | 42 (36.2) | 14 (12.1) | 0.606 |
| No | 113 (49.3) | 65 (57.5) | 34 (30.1) | 14 (12.4) | |
| Smoke inside house * | | | | | |
| Yes | 112 (49.1) | 54 (48.2) | 42 (37.5) | 16 (14.3) | 0.143 |
| No | 116 (50.9) | 71 (61.2) | 33 (28.4) | 12 (10.4) | |

* Total n is not equal to n = 233 for these variables due to missing values. Overall column displays the column percentages (%) of column totals in parentheses and OSA subcategories columns display row percentages (%) of the row totals in parentheses.

Table 2 demonstrates an ordinal logistic regression univariable model (unadjusted) which showed significant associations between high values of AHI and obesity and loud snoring. In addition, the ordinal logistic multiple regression model (adjusted) revealed that the odds of having a high AHI (mild to severe vs. normal) was 2.48 (95% CI, 1.35 to 11.90) higher for males compared to females which was statistically significant, $p = 0.004$. For obese individuals, the odds of having high AHI (mild to severe vs. normal) was 3.89 (95% CI, 2.10 to 49.02, $p < 0.0001$) times than that of non-obese participants. Participants who snored loudly were 2.14 (95% CI, 1.16 to 8.56) times the odds of those who did not snore loudly to have a high AHI (mild to severe vs. normal), which was statistically significant, $p = 0.016$. The test of parallel lines (Chi-Square test statistics = 7.176, degrees of freedom = 7; $p = 0.411$) satisfied the proportional odds assumption of the ordinal regression model. Interactions between potential effect modifiers were examined but none were significant.

**Table 2.** Ordinal Regression * analysis of OSA and selected risk factors (n = 233).

| Variable | OR$_{unadj}$ (95% CI) | OR$_{adj}$ (95% CI) | *p* Value |
|---|---|---|---|
| Sex | | | |
| Male | 1.60 (0.96, 2.64) | 2.48 (1.35, 11.90) | 0.004 |
| Female | 1.00 | 1.00 | |
| Age group, in years | | | |
| 50 and older | 1.12 (0.58, 2.18) | 1.33 (0.62, 3.77) | 0.463 |
| 40–49 | 1.63 (0.76, 3.49) | 1.20 (0.52, 3.31) | 0.674 |
| 30–39 | 1.34 (0.69, 2.59) | 1.02 (0.48, 2.77) | 0.964 |
| 18–29 | 1.00 | 1.00 | |
| Body Mass Index | | | |
| Obese | 1.30 (1.69, 4.77) | 3.89 (2.10, 49.02) | <0.0001 |
| Other | 1.00 | 1.00 | |
| Excessive Daytime Sleepiness | | | |
| Yes | 1.47 (0.76, 2.84) | 1.50 (0.72, 4.47) | 0.281 |
| No | 1.00 | 1.00 | |
| Loud Snoring | | | |
| Yes | 2.68 (1.51, 14.40) | 2.14 (1.16, 8.56) | 0.016 |
| No | 1.00 | 1.00 | |

* ORs derived from ordinal regression model with severity of OSA using AHI (the severity of OSA is classified as follows: none/normal—AHI < 5 per hour; mild—AHI $\geq$ 5, but <15 per hour; moderate to severe —AHI $\geq$ 15 per hour).

## 3. Discussion

The global population estimated prevalence of OSA ranges from 7.8% to 77.2% for OSA defined by AHI $\geq$ 5/h and from 4.8% to 36.6% for moderate-severe OSA (AHI $\geq$ 15/h) among the population aged 30–69 years [33]. The prevalence of OSA diagnosed with AHI $\geq$ 5/h among Australian Aboriginal people was 88% and of those, 52% were males and 48% were females [34]. A recent study among Australian Aboriginal and Torres Straight Islanders (ATSI) peoples reported that among ATSI, the prevalence of OSA diagnosed with AHI > 5/h was 88.5%, and of those, 46% were males and 42% were females [35]. A comparable study of Indigenous people from the United States reported that for American Indians, OSA prevalence was 56% for AHI $\geq$ 5/h and 23% for AHI $\geq$ 15/h [15]. The prevalence of OSA reported in this study was 45.0%, with moderate to severe OSA of 12.1% among the participants in Community A. The prevalence of OSA is 51.0% for males and 41.0% for females. The prevalence of OSA in this study is lower than reported for Indigenous peoples elsewhere and within the percentages for the global population. However, the prevalence of OSA in this study was higher than the national statistics in Canada. The Canadian Health Measures Survey (CHMS) in 2016 and 2017 reported that the prevalence of sleep apnea, having been diagnosed by a health care professional, was 6.4% among adults. In addition, nearly one-third (30%) of Canadian adults were considered to be at intermediate or high risk for sleep apnea based on the STOP-Bang tool [2].

This community-based research study regarding sleep revealed that the three main factors associated with severity of OSA were sex, obesity, and loud snoring. None of the socio-economic factors were significantly associated with OSA. Similar to previous studies [36–43], OSA was less prevalent in females than in males in this study. Many studies have discussed the sex differences in the dimensions, structure, and physiological behavior of the upper airways [36,40,44,45] as the reason for this difference. Upper airway length is typically longer in males than in females and related to high airway collapsibility [36,44]. Upper airways are less collapsible and stable during sleep in females due to the involvement of female sex hormones [36], which appear to provide a protective effect. Further, Lin et al. suggested that because women typically have a shorter oropharynx than men, they have a lower risk of collapsibility [40]. These authors added the observation that having a longer, floppy oropharynx and larger, fatter, posterior tongue are related to the worsened OSA

conditions in men [40]. Upper airway soft tissue structures appear to be larger in men compared to women, and these differences in anatomy between men and women tend to increase the risk of airway collapse in men during sleep [45]. Finally, sex differences in blood carbon dioxide levels, arousal response to episodes of apnea, and sleep architecture conjointly may explain the differences in prevalence and severity of OSA between males and females [46].

The association between obesity and OSA has been well studied [40,46–51]. Higher body mass index (BMI) has been associated with a greater severity of OSA for both sexes [47], and similar results were found in this study. Gami et al. reported that ~70% individuals with OSA were obese and the prevalence of OSA in men and women with obesity was about 40% [50]. The pathophysiology of OSA in people with obesity is also affected by fat distribution. Numerous studies have shown that OSA is highly correlated with waist circumference, upper body skin-fold thickness, and visceral fat content [50,52–57]. In addition, it has also been shown that there is a greater tendency for visceral and hepatic fat accumulation to occur in males than females [37]. The extra body fat distribution in the trunk, head, and neck makes people more prone to OSA due to increased deposition of fat around the airways [40]. In this regard, a higher prevalence of OSA is reported in males than females [40,58]. The posterior tongue contains about 30% fat, and with weight gain, tongue weight and the percentage of fat increases. In this way, fat deposition in the upper airway and in the posterior tongue affects the pathogenesis of OSA [40]. Another study reported that fat distribution around the thorax may increase the oxygen demand due to reduction of chest compliance and functional residual capacity [51].

One of the reasons for the high OSA prevalence found in this study could be due to high rates of obesity in this First Nation community. The prevalence of obesity overall (49.4%) in this cohort is higher than the national average (36%) for on-reserve adults (18+ years) in Canada [59]. The prevalence of obesity is 38% for males and 58% for females. Previous studies have reported high rates of obesity in Canadian Indigenous peoples which include First Nations [60–62]. Katzmarzyk et al. reported that the prevalence of obesity among Canadian First Nations was high, ranging from 29% in youth (male and female) to 60% in adult females [62]. First Nation females in Canada have greater subcutaneous fat than do their male counterparts (sum of skinfolds). First Nation adult females also have a greater mean body mass, BMI, trunk-to-extremity skinfold ratio, and sum of skinfold than females of European ancestry. Among adult males, First Nation males have a greater BMI and sum of skinfold than males of European ancestry. Lear et al. reported that Indigenous men and women had a larger waist circumference and waist-to-hip ratio compared to men and women of European ancestry, but there was no difference in body mass index between the two groups [60]. Indigenous men also had a higher percent body fat and subcutaneous abdominal adipose tissue than men of European ancestry. Body mass index and waist circumference were highly correlated with measures of total and central adiposity for men and women of both ethnic groups (Indigenous and European) [60]. Another study among women reported that obesity was more common in First Nations women (48.1%) than in women with European ancestry (36.2%) [61]. In addition, the same authors reported that First Nations women differed from women with European ancestry in terms of fat and lean tissue mass and fat distribution [61].

Snoring is a common symptom of potential sleep disordered breathing [63]. Louder snoring is associated with more severe OSA [64]. O'Connor et al. showed there was an increased risk of snoring in Native American Indian women [13]. Other studies have also shown an increased risk of snoring among Indigenous peoples compared to non-Indigenous people [15,16]. In this study, the prevalence of loud snoring was 28.8% and loud snoring was significantly associated with an increased risk of severity of OSA.

*Strengths and Limitations*

Strengths of this study included the large number of participants and the inclusion of a number of potential factors including lifestyle, socio-demographic, and sleep characteristics.

This study was one of the first to examine obstructive sleep apnea (OSA) in adults living in a rural Cree First Nation community in Saskatchewan, Canada using a Level 3 Home Sleep Test. The advantage of the Level 3 test was that it provided the measurements of airflow, snoring, respiratory excursion, body position, heart rate, and oxygen saturation. However, some limitations should also be noted. Level 3 studies do not record sleep; therefore, severity of OSA is estimated using the respiratory event index, which is the number of desaturation events per hour of total recording time [3]. The respiratory event index may underestimate AHI because it measures time when the patient is not actually asleep and does not detect arousals from sleep [3].

Comparison of AHI from the home-based ApneaLink test and sleep laboratory-based tests demonstrated that four or more hours of recording time has provided the most accurate testing and suggest that a recording time of less than four hours may lead to more-frequent false negative tests [65]. In this study, some of the AHI evaluation recording times were less than four hours as these tests were self-monitored but the mean AHI values were not significantly different from those that were four hours and greater ($p > 0.05$). There were a few other concerns such as not wearing the equipment during the entire night, returning the equipment on time, and repeating the test if the information was not recorded.

The questionnaire survey data were self-reported, with possible recall bias. The willingness to participate in a Level 3 one-night home sleep test was 56% (233/418) of those who completed the survey questionnaire. However, the participants in this sample were volunteered for follow-up. Therefore, this is not a random sample of the population. An additional consideration in this study was that the population was young (mean age 36 years (SD = ±14 years) for males and 40 years (SD = ±15 years) for females). Thus, the population in this study included a largely population of pre-menopausal population among the females; and it is well known that pre-menopausal females have a lower prevalence of OSA than do males in the same age group. Although we found associations between several factors and OSA, causal relations could not be assessed due to the cross-sectional nature of the data. Although there are some similarities, Canadian Cree First Nations are different from any other Indigenous groups, which may explain the deviation in findings of prevalence of OSA from the other Indigenous groups.

## 4. Methods

### 4.1. Study Sample

The baseline survey of the First Nations Sleep Health Project (FNSHP) was completed between 2018 and 2019 in collaboration with two Cree First Nation communities (Community A and Community B) in Saskatchewan, Canada. The overall goal of the FNSHP was to study the relationships between sleep disorders, risk factors, and comorbidities, and to evaluate local diagnoses and treatments. This project was co-developed with two communities with discussion and support from Elders, leaders, health care workers, and youth in each community. The current research has been developed after an eight-year relationship with the two communities (two years of learning and preparing (ten meetings and six years conducting a previous Canadian Institutes of Health Research (CIHR) project on housing and respiratory health [29]). Elders and community leadership have been involved in all aspects of the current project which has been co-developed through numerous meetings, sharing circles and team participation at community events such as Treaty Days as we gradually learned how to work together. The findings/results from the first collaboration [29] led to the current collaboration on sleep and health. The project was intended to provide benefits for the communities, both in terms of knowledge development, as well as in human resources and capacity strengthening. Approval was obtained for the current sleep project in each of the communities from every level of governance in the community. There was an opportunity for everyone in the communities to participate. The research team members were informed about the customs and cultural ceremonies of practice in each of the two communities through the work of the previous project, as well as through an ongoing partnership that continues to occur in the current project. Both communities had

co-developed and signed Research Agreements prior to the application being submitted to CIHR for funding. Subsequently, an Operational Agreement was signed with Community A for the use of space and the payment of research staff. The research was carried out collaboratively. The survey was developed with input from the communities to ensure sensitivity to customs, culture and linguistic factors. In addition, the communities reviewed and edited the surveys before they were submitted to the University of Saskatchewan's Bio-medical Research Ethics Board for approval. A Certificate of Approval was obtained from the University of Saskatchewan's Biomedical Research Ethics Board (Certificate No. Bio #18-110). In addition, adherence to Chapter 9 (Research Involving the First Nations, Inuit, & Métis Peoples of Canada) in the Tri-Council Policy Statement: Ethical Conduct for Research Involving Humans was undertaken [66]. The data generated are co-owned by the communities, the interpretation of findings/results requires active involvement of community participants, and the dissemination of results requires review and approval by the communities prior to submission. Thus, we obtain approval from community members before submitting any abstracts/reports/manuscripts for presentation or publication. This project supports capacity building through hiring and training local community members and summer students. Elders/Knowledge Keepers from each of the partner communities are participating as active collaborators in this research project. This project respects both individuals' rights to privacy and confidentiality, as well as the rights of the communities involved. As a result, individual participants provided written consent to participate in this research collaboration.

### 4.2. Data Collection

Research Assistants were hired from the communities and trained to conduct the baseline surveys in their respective community. Adults 18 years and older were invited to participate in the baseline survey. Data were collected via interviewer-administered questionnaires (in Community A & B) and clinical assessments (only conducted in Community A). The survey collected information on demographic variables, individual and contextual determinants of sleep health. Objective clinical measurements included a Level 3 Home Overnight Sleep Test and actigraphy. This manuscript is based on data from the questionnaires and the one-night at-home sleep tests collected in Community A.

### 4.2.1. Measurements

Anthropometric measurements (abdominal girth, neck circumference, hip circumference, height, and weight) were obtained. Height was measured against a wall using a fixed tape measure with participants standing in stockings on a hard floor. Weight was measured using a calibrated spring scale with participants dressed in indoor clothing and taking off their shoes. Using clinical measures of weight and height, body mass index (BMI) was calculated based on the equation of BMI = weight (kg)/(height (m))$^2$ [67]. In addition, an oxygen saturation was measured using a Pulse Oximeter (Contec Medical Systems, Qinhuangdao, Hebei Province, China) and two blood pressure measurements taken using the LifeSource$^{(R)}$ digital blood pressure monitor (A&D Medical, San Jose, CA, USA) were recorded.

Level 3 home sleep assessments were obtained by sending Level 3 overnight sleep test kits home with instructions for participants who presented for clinical measurements. Testing was conducted using the ApneaLink Air$^{TM}$ ResMed system (ResMed, San Diego, CA, USA). Continuous variables derived from the Level 3 tests were: the apnea/hypopnea index (AHI); oxygen desaturation index; lowest and average oxygen saturation; cumulative duration of oxygen saturation $\leq 88\%$; obstructive index; central index; as well as apneas; hypopneas; and snoring counts.

Trained Research Assistants prepared the ApneaLink Air device connecting all accessories (nasal cannula, effect sensor, oximeter, and belt) before giving it to the participants. They then fit the belt to the participant and showed the participant how to position the device over the center of their chest comfortably and how to position the oximeter com-

fortably on a finger. In addition, they showed the participants how to insert the nasal cannula securely and comfortably. Medical tape was provided to anchor the cannula onto the cheeks of the participants. Once the device was properly set up, instructions were provided on how to start the test, how to stop the test, and how to disassemble and return the device to the clinic on the following morning. A Participant's Instruction Sheet was given to each participant to take home in case they forgot how to wear the equipment. Once the kit was returned, the research assistants downloaded the results and checked to see if the test had been properly recorded. Upon successful completion of the test, the participant was provided a \$50 honorarium for completing the survey questionnaire and the one-night home sleep test.

### 4.2.2. Definitions

In adherence to the American Academy of Sleep Medicine (AASM)'s Position Statement, obstructive apnea is defined as a reduction in airflow of 90% in the presence of thoracoabdominal movements for a period of at least 10 s [68]. In addition, obstructive hypopnea was defined as a decrease of 30% or more in the sum of thoracoabdominal movements for at least 10 s associated with a decrease in oxygen saturation of at least 3% [68,69]. The Apnea Hypopnea Index (AHI), the number of apneic and hypopneic events per hour of monitoring time, was used to indicate the severity of OSA. It is generally expressed as the number of events per hour. Based on the AHI, the severity of OSA was classified as follows: none/normal—AHI < 5 per hour; mild OSA—AHI $\geq$ 5, but <15 per hour; moderate OSA—AHI $\geq$ 15, but <30 per hour; severe OSA—AHI $\geq$ 30 per hour [70]. For the analysis, moderate and severe OSA groups were combined due to the small number of cases in the severe OSA group.

### 4.2.3. Variable Descriptions

The questionnaire collected the following demographic variables: age; sex; engaged in non-traditional use of tobacco (cigarette smoking); and money left at the end of the month (socioeconomic status); and housing environmental conditions such as damage caused by dampness, moldy or musty smell, signs of mold in house, and cigarette smoking inside the house. Loud snoring was obtained using the question, 'Do you snore loudly (louder than talking or loud enough to be heard through closed doors)?' Crowding index was reported as the ratio of the number of people in the house to the number of bedrooms which was categorized into two groups >1 and $\leq$1. Excessive daytime sleepiness was defined by the Epworth Sleepiness Scale (ESS) score of 11 to 24 as 'yes'. Normal, overweight, and obese were defined as BMI $\leq$ 25 Kg/m$^2$; BMI = 25–29.9 Kg/m$^2$ and BMI $\geq$ 30 Kg/m$^2$ respectively.

### 4.3. Statistical Analysis

Statistical analyses were conducted using Statistical Package for the Social Sciences (SPSS) software (IBM Corp. Released 2020. IBM SPSS Statistics for Windows, Version 27.0. Armonk, NY, USA: IBM Corp.). Descriptive statistics, mean and standard deviation (SD) were reported for continuous variables. For categorical variables, frequency and percentage (%) were reported. Chi-square tests were used to determine the bivariable association of severity of OSA with the independent variables of interest. Ordinal logistic regression models [71,72] in SPSS PLUM procedure were used to predict the relationship between severity of OSA (normal (AHI < 5), mild OSA (AHI 5–14.99) and moderate to severe OSA (AHI $\geq$ 15)) and a set of explanatory variables. A series of ordinal logistic regression models were fitted to determine whether potential risk factors, confounders, and interactive effects contributed significantly to the severity of OSA. Based on bivariable analysis, variables with $p < 0.20$ were included in the multivariate model. All variables that were statistically significant ($p < 0.05$), as well as important clinical factors (sex, age, obese, excessive daytime sleepiness, and snoring), were retained in the final multivariable model. Interactions between potential effect modifiers were examined and were also retained in the final model if the $p$-value was <0.05. The strength of association was presented by odds

ratio (OR) and 95% confidence interval (CI) [73]. The test of parallel lines in SPSS was used to test the proportional odds assumption for ordinal regression model [71,72].

## 5. Conclusions

This is the first study of OSA that engaged First Nation peoples in all aspects of the project. In summary, the prevalence and the severity of OSA were high in one First Nation community compared to the national average but were within the global estimates of OSA. From this study, male sex, obesity, and loud snoring were the three main factors associated with the severity of OSA. OSA treatment and prevention in these communities should align with understanding the impact that social determinants have on First Nations peoples, as well as other risk factors for OSA.

**Author Contributions:** Conceptualization, J.A.D., S.A., M.K., P.P., D.C.R., S.K., N.K., M.F., R.S.; Data curation, B.P.R., K.M.; Formal analysis, C.P.K.; Funding acquisition, J.A.D., S.A., M.K. and P.P.; Investigation, J.A.D., S.A., M.F., M.K. and P.P.; Methodology, J.A.D., P.P., S.A., M.K., C.P.K., M.F.; Project administration, P.P.; Resources, J.S., R.S., M.F.; Supervision, J.A.D. and P.P.; Visualization, S.A., J.S. and V.R.R.; Writing—original draft, J.A.D., C.P.K., P.P.; Writing—review & editing, J.A.D., C.P.K., B.P.R., K.M., S.A., D.C.R., S.K., N.K., J.S., V.R.R., M.F., M.K. and P.P. All authors have read and agreed to the published version of the manuscript.

**Funding:** This research was funded by a grant from the Canadian Institutes of Health Research, "Assess, Redress, Re-assess: Addressing Disparities in Sleep Health among First Nations People," CIHR MOP-391913-IRH-CCAA-11829-FRN PJT-156149.

**Institutional Review Board Statement:** The study was conducted according to the guidelines of the Declaration of Helsinki and approved by the Biomedical Research Ethics Board of University of Saskatchewan (Bio #18-110 and date of approval: 21 June 2018).

**Informed Consent Statement:** Written informed consent has been obtained from all participants involved in the study.

**Data Availability Statement:** The summarized data presented in this study are available on request from the corresponding author. The data are not publicly available due to the agreement with two participating communities.

**Acknowledgments:** The First Nations Sleep Health Project Team consists of: James A Dosman, (Designated Principal Investigator, University of Saskatchewan, Saskatoon, SK Canada); Punam Pahwa, (Co-Principal Investigator, University of Saskatchewan, Saskatoon SK Canada); Malcolm King, (Co-Principal Investigator, University of Saskatchewan, Saskatoon, SK Canada), Sylvia Abonyi, (Co-Principal Investigator, University of Saskatchewan, Saskatoon, SK Canada); Co-Investigators: Mark Fenton, Chandima P Karunanayake, Shelley Kirychuk, Niels Koehncke, Joshua Lawson, Robert Skomro, Donna Rennie, Darryl Adamko; Collaborators: Roland Dyck, Thomas Smith-Windsor, Kathleen McMullin; Rachana Bodani; John Gjerve; Bonnie Janzen; Vivian R Ramsden; Gregory Marchildon; and Kevin Colleaux; Project Manager: Brooke Russell; Community Partners: Jeremy Seeseequasis; Clifford Bird; Roy Petit; Edward Henderson; Raina Henderson; Dinesh Khadka. We are grateful for the contributions from Elders and the community leaders who facilitated the engagement necessary for the study, research assistants who worked and all participants who engaged in this study.

**Conflicts of Interest:** The authors declare that they have no known competing financial interests or personal relationships that could have appeared to influence the work reported in this study. The funders had no role in the design of the study; in the collection, analyses, or interpretation of data; in the writing of the manuscript, or in the decision to publish the results.

## Abbreviations

| OSA | Obstructive sleep apnea |
| --- | --- |
| AHI | Apnea–hypopnea index |
| BMI | Body mass index |
| USA | United States of America |
| NIHB | Non-Insured Health Benefits |

CPAP     Continuous Positive Airway Pressure therapy
FNSHP    First Nations Sleep Health Project
CIHR     Canadian Institutes of Health Research
AASM    American Academy of Sleep Medicine
ESS      Epworth Sleepiness Scale
SPSS     Statistical Package for the Social Sciences
SD       Standard Deviation
OR       Odds ratio
CI        Confidence interval
ATSI      Aboriginal and Torres Straight Islanders
CHMS    Canadian Health Measures Survey

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
