# Peer review of "Obesity, Sex, Snoring and Severity of OSA in a First Nation Community in Saskatchewan, Canada"

_2624-5175, doi:10.3390/clockssleep4010011_

Round 1
Reviewer 1 Report
Summary
In this study, Dosman et al. investigated potential risk factors that may be related to obstructive sleep apnea (OSA) in the Indigenous people living in a First Nation community in Saskatchwan, Canada. The authors performed an interviewer-administered survey and a level 3 at-home sleep test to study the relationship between OSA and personal medical conditions, as well as socioeconomic factors. They found that the prevalence of OSA is higher than the average national number in Canada, but lower compared to the data from Indigenous people living in Australia or United States. Further statistical analysis revealed that sex, obesity, and snoring are three major risk factors for OSA in the First Nation community Indigenous people, while socioeconomic factors such as living area and housing condition are not associated with the OSA incidence rate.
The main contribution of this manuscript is that it is the first study to investigate the risk factors of OSA in Indigenous peoples living in Canada. The dataset collected by the authors and their findings will appeal to both sleep community and researchers interested in public health. However, similar studies focused on First peoples in other countries have been published (as in ref 15, 41 in the manuscript). The experimental design and methodology applied in this paper are reasonable but not novel. More importantly, the manuscript contains quite a lot of grammatical, spelling and formatting mistakes. I listed a few examples below, and I highly recommend the authors seek for assistance from a professional proofreader or colleagues who are native English speakers. As a conclusion, I’m afraid that the manuscript is not acceptable for the journal in its current form. Below, I provide some suggestions that I think would be helpful this paper for resubmission.
Major concerns:
- In table 1, some key information that describe these factors were missing. For example, how to define obese participants in this survey, BMI>30?
- In table 3, only certain factors (sex, age, obesity, and snoring) in table 2 were chosen for further analysis. It’s not clear to me why the authors only pick these factors.
- Here are some notes for grammar and formatting mistakes in the paper:
- Line 51: an extra space between “was” and “related”.
- Line 52: “co-morbidities” should be “comorbidites”. “amongst” should be replaced with “among”.
- Line 58: reference citation should be inserted at the end of this sentence.
- Line 68: replace “peoples” to “people”.
- Line 74: replace “conditions” to “condition”.
- Line 88: replace “in-home” to “at-home”.
- Line 116: replace “are” to “were”.
- Line 119: replace “have” to “had”.
- Line 164: replace “participant” to “participants”.
- Line250: an extra space between “of” and “OSA”.
- Line 259: it’s not clear to me whether 88.5% is for ATSI people or non-ATSI people, please provide more information.
Minor comment:
- Suggestion: data presentation is acceptable by using tables, but data visualization using charts will be helpful for data interpretation.
Author Response
Review Report Form #1
Comment #1
The main contribution of manuscript is that it is the first study to investigate the risk factors of OSA in Indigenous peoples living in Canada. The dataset collected by the authors and their findings will appeal to both sleep community and researchers interested in public health. However, similar studies focused on First peoples in other countries have been published (as in ref 15, 41 in the manuscript). The experimental design and methodology applied in this paper are reasonable but not novel. More importantly, the manuscript contains quite a lot of grammatical, spelling and formatting mistakes. I listed a few examples below, and I highly recommend the authors seek for assistance from a professional proofreader or colleagues who are native English speakers. As a conclusion, I’m afraid that the manuscript is not acceptable for the journal in its current form. Below, I provide some suggestions that I think would be helpful this paper for resubmission.
Response: Thanks for the comment and the manuscript was proofread by a Canadian Native English speaker.
Major concerns:
Comment #2: In table 1, some key information that describe these factors were missing. For example, how to define obese participants in this survey, BMI>30?
Response: Thank you for the comment. We have included a detailed variable description section. Please see section 2.2.3.
Comment #3: In table 3, only certain factors (sex, age, obesity, and snoring) in table 2 were chosen for further analysis. It’s not clear to me why the authors only pick these factors.
Response: As described in Section 2.3 Statistical Analysis (Lines 198-204), in the regression model building strategy, we have chosen to initially include in the multivariable modelling those variables with a significance level of p<0.20. All variables that were statistically significant (p < 0.05), as well as important clinical factors (sex, age, obesity, excessive daytime sleepiness and snoring), were retained in the final multivariable model.
Comment #4:
- Here are some notes for grammar and formatting mistakes in the paper:
- Line 51: an extra space between “was” and “related”.
Response: Thank you. Extra space was deleted.
- Line 52: “co-morbidities” should be “comorbidites”. “amongst” should be replaced with “among”.
Response: The word “co-morbidities” is corrected through out the manuscript. The word ‘amongst’ is replaced with ‘among’.
- Line 58: reference citation should be inserted at the end of this sentence.
Response: Reference citation is inserted at the end of the sentence.
- Line 68: replace “peoples” to “people”.
Response: The word ‘peoples’ is replaced with ‘people’. However, in Canada, due to the number of First Nations, e. g. There are 74 sovereign First Nations in Saskatchewan; thus, it is not one people but many peoples.
- Line 74: replace “conditions” to “condition”.
Response: The word ‘conditions’ is replaced with ‘condition’.
- Line 88: replace “in-home” to “at-home”.
Response: The word ‘in-home’ is replaced with ‘at-home’.
- Line 116: replace “are” to “were”.
Response: The word ‘are’ replaced with ‘were’.
- Line 119: replace “have” to “had”.
Response: The word ‘have’ replaced with ‘had’.
- Line 164: replace “participant” to “participants”.
Response: The word ‘participant’ replaced with ‘participants.
- Line250: an extra space between “of” and “OSA”.
Response: Thank you. Extra space is deleted.
- Line 259: it’s not clear to me whether 88.5% is for ATSI people or non-ATSI people, please provide more information.
Response: The prevalence of OSA was 88.5% among ATSI people. This sentence is modified to following:
“A recent study among Australian Aboriginal and Torres Straight Islanders (ATSI) peoples reported that among ATSI the prevalence of OSA diagnosed with AHI>5/hour was 88.5%, and of those 46% were males and 42% were females [41].”
Minor comment:
- Suggestion: data presentation is acceptable by using tables, but data visualization using charts will be helpful for data interpretation.
Response: As requested by first reviewer, Table 1 is converted to a Figure 1 and old Table 1 is removed. Other two tables are labelled as Table 1 & Table 2.
Reviewer 2 Report
I reviewed the manuscript "Obesity, sex, snoring, and severity of OSA in a First Nation community in Saskatchewan, Canada," which appears to be interesting. At the moment, I have some questions and suggestions about the manuscript, which are listed below.
1. Why did the authors leave out the data from Community B? If it isn't justified, why do they bring it up to the authors of this article?
2. Lines 211-212, "The mean apnea link evaluation......minutes," belongs more in the method section than in the result section.
3. Is there a specific reason why the authors chose to present the parentage data comparing different grades of OSA for the total number of individuals for that grade, e.g. Normal OSA, etc., rather than sub-variables in Table 2? For example, for different OSA categories, the percentage date is calculated by dividing the total number of individuals categorized in that category by the total number of sub-variables, such as male or female in sex. I strongly believe that calculating the percentage of OSAs against the total number of sub-variables of variables listed in the table would have made more sense.
4. In line 265 it is stated that "the prevalence of OSA in this study is lower than that reported for Indigenous peoples elsewhere and within the percentages for the global population." Is there any kind of explanation for this kind of deviation in finding?
Author Response
Review Report Form #2
Comments and Suggestions for Authors
I reviewed the manuscript "Obesity, sex, snoring, and severity of OSA in a First Nation community in Saskatchewan, Canada," which appears to be interesting. At the moment, I have some questions and suggestions about the manuscript, which are listed below.
Comment #1
1. Why did the authors leave out the data from Community B? If it isn't justified, why do they bring it up to the authors of this article?
Response: We collected survey questionnaire data and one-night sleep test data from July 2018-November 2019 in Community A. Then we started collecting survey questionnaire data from Community B in December 2019. We had to discontinue data collection in Feb 2020 due to COVID 19 situation. We were not able to go back to either First Nation Community as the risk of spreading COVID 19 during 2020 and 2021. We have included a statement to clearly indicate that in Section 2.2, this manuscript was based on data from Community A.
Comment #2
2. Lines 211-212, "The mean apnea link evaluation......minutes," belongs more in the method section than in the result section.
Response: Thank you for the comment. The Apena Link evaluation was described in the methods section. This sentence is the mean value for the duration that the Apnea Link data was collected, and as such belongs in the results section. We have corrected the sentence to more clearly delineate that this was a mean measure.
Comment #3
3. Is there a specific reason why the authors chose to present the parentage data comparing different grades of OSA for the total number of individuals for that grade, e.g. Normal OSA, etc., rather than sub-variables in Table 2? For example, for different OSA categories, the percentage date is calculated by dividing the total number of individuals categorized in that category by the total number of sub-variables, such as male or female in sex. I strongly believe that calculating the percentage of OSAs against the total number of sub-variables of variables listed in the table would have made more sense.
Response: The table 2 presented with column percentage considering the totals from different grades of OSA. As reviewers requested Table 2 (now Table 1) presents with row percentages considering the totals for the sub-categories of variables.
Comment #4
4. In line 265 it is stated that "the prevalence of OSA in this study is lower than that reported for Indigenous peoples elsewhere and within the percentages for the global population." Is there any kind of explanation for this kind of deviation in finding?
Response: To our knowledge, this is the first study reporting information of this kind for Canadian First Nation peoples. We can only speculate that although there are some similarities, Canadian Cree First Nations are different in this respect from any other Indigenous groups.

Round 2
Reviewer 1 Report
The authors have properly addressed all the points that I have raised on the previous version of the manuscript, and I don’t have further comments except the following one.
A minor comment:
For Figure1, the figure legends are too small to see, either make them bigger or use different colors for different columns.
Author Response
Comments and Suggestions for Authors
Comment #1:
The authors have properly addressed all the points that I have raised on the previous version of the manuscript, and I don’t have further comments except the following one.
Response: Thank you!
Comment #2:
A minor comment:
For Figure1, the figure legends are too small to see, either make them bigger or use different colors for different columns.
Response: As reviewer’s suggestion, Figure 1 was created with colour for different columns.
Reviewer 2 Report
The article got improved following the reviewer's comments!
Author Response
Comments and Suggestions for Authors
Comment #1:
The article got improved following the reviewer's comments!
Response: Thank you!